# The predicting formula and scoring system for cardiac iron overload for thalassaemia children: Study from a middle-income country

**Syarif Rohimi**[1]*, **Bambang Budi Siswanto**[2], **Muchtaruddin Mansyur**[3],
**Djajadiman Gatot**[4], **Ina Sutanto**[5], **Jacub Pandelaki**[6], **Amiliana M. Soesanto**[2],
**Teddy Ontoseno**[7]

**1** Rumah Sakit Anak dan Bunda RSAB Harapan Kita, Jakarta, Indonesia, **2** Harapan Kita National Cardiac Centre Hospital, Jakarta, Indonesia, **3** Department of Community Medicine, Faculty of Medicine Universitas Indonesia, Jakarta, Indonesia, **4** Department of Paediatrics, Faculty of Medicine Universitas Indonesia, Jakarta, Indonesia, **5** Department of Clinical Pathology, Faculty of Medicine Universitas Indonesia, Jakarta, Indonesia, **6** Department of Radiology, Faculty of Medicine Universitas Indonesia, Jakarta, Indonesia, **7** Department of Paediatrics, Faculty of Medicine Universitas Airlangga, Jakarta, Indonesia

* syarohmi@yahoo.com

## Abstract

Magnetic resonance imaging T2* screening is the gold standard for detecting cardiac iron overload in thalassemia, but its implementation in Indonesia is limited by the high costs. A predicting formula and scoring system based on low-cost investigations is needed. This cross-sectional study was conducted among thalassemia aged 6–18 years at Rumah Sakit Anak dan Bunda RSAB Harapan Kita Indonesia, during October 2017 to April 2019. All subjects were scheduled for clinical examination, laboratory tests, ECGs, echocardiography, tissue Doppler imaging, and MRIT2*. Multivariate logistic regression was used to identify the formula, simplifying to a scoring system, and risk classification for myocardial iron overload using odds ratio (OR) and 95% confidence interval (CI). Significance was set as $p < 0,05$. We recruited 80 children, of those, 8 (10%) were classified as cardiac iron overload based on MRI T2* screening. Multivariate logistic regression showed determinant factors for cardiac iron overload were hemoglobin (95% CI:1.92–369.14), reticulocyte (95% CI:1.14–232.33), mitral deceleration time (DT) (95% CI:1.80–810.62,), and tricuspid regurgitation (TR Vmax) (95% CI:1.87–1942.56) with aOR of 26.65, 14.27, 38.22, and 60.27 respectively. The formula for cardiac iron overload was decided as 9.32 + 3.28 (Hb) + 2.9 (reticulocyte) + 3.64 (DT) + 4.1 (TR Vmax). A scoring system was defined by simplifying the formula of Hb ≤ 8.2 g/L, reticulocyte ≤0.33%, DT ≤ 114.5 cm/s, and TR Vmax ≥ 2.37 m/s were given a score of 1, while others were assigned 0. Total scores of 0 or 1, 2 and 3 or 4 were categorized as low, moderate, and high risk for iron cardiac overload. The cardiac iron overload formula was 9.32 + 3.28 (Hb) + 2.9 (reticulocyte) + 3.64 (DT) + 4.1 (TR Vmax). Variables of Hb ≤ 8.2 g/L, reticulocyte ≤0.33%, DT ≤ 114.5 cm/s, and TR Vmax ≥ 2.37 m/s were given a score of 1, while others were assigned 0. Total scores of 0 or 1, 2, and 3 or 4 were categorized as low, moderate, and high risk for iron cardiac overload.

**Data Availability Statement:** All relevant data are within the manuscript and its Supporting Information files.

**Funding:** The author(s) received no specific funding for this work.

**Competing interests:** The authors have declared that no competing interests exist.

## Introduction

Thalassaemia is a prevalent genetic disorder, with an incidence rate of 1.4–5 per 1,000 live births. Furthermore, its carriers make up 3% of the global population, with 60,000 symptomatic births occurring every year [1]. Thalassaemia β major is known to cause severe clinical symptoms and morbidity, as well as lead to death [2, 3]. In Indonesia, there has been an increase in the number of thalassaemia patients. In recent years, 9,009 (85.7%) out of 10,515 patients were aged 0–20 years in 2019 [4].

Regular blood transfusions are required to treat thalassemia and might cause iron overload, organ dysfunction, heart failure (HF), cardiac fibrosis, pulmonary hypertension and death [1, 3, 5]. Early detection of cardiac iron overload using magnetic resonance imaging (MRI) T2* can prevent heart failure and death [5]. However, at present, MRI T2* is still limited in Indonesia due to its high cost as well as the lack of scanners required. Therefore, this study aims to develop a formula and scoring system for cardiac iron overload to improve management, prevent death, and increase life expectancy based on low-cost investigations. As far as we are aware, there are no study regarding formula and scoring system on cardiac iron overload for thalassaemia major children in Indonesia.

## Materials and methods

This cross sectional study was carried out at Rumah Sakit Anak dan Bunda Harapan Kita, Indonesia, during October 2017 to April 2019. Thalassemia major aged 6–18 years with ferritin levels > 1,000 ng/mL were included. The diagnosis of thalassemia major was based on Hb analysis. The clinical and demographic data included age, sex, age at first diagnosis and chelation, interval between transfusions, and type of chelation therapy were collected. Patients with congenital heart disease or complications unrelated to thalassemia were excluded. Chelation adherence was defined based on the ratio of total drug ingested and drug prescribe per month within the last three months. Nutritional status was determined based on left arm circumference.

All subjects were scheduled for laboratory tests, ECG, left and right cardiac function, tricuspid valve regurgitation (TR Vmax), basal left ventricular and septal myocardial velocity with tissue doppler imaging (TDI) one week before blood transfusion. The laboratory test consisted of hemoglobin (Hb), reticulocyte, immature granulocyte (IG), WBC, platelets, urea, creatinine, AST, ALT, SI, and TIBC measurements [6]. The ECG results were manually assessed. Echocardiography and TDI used Epiq7 and were done according to American Society of Echocardiography and a clinician's guide to TDI [7, 8]. Echocardiograhy and TDI variables were done 3 time and calculated means. MRI T2* was performed at the Department of Radiology, Cipto Mangunkusumo Hospital and Premier Hospital Jakarta. MRI T2* < 20 ms was classified as cardiac iron overload. All authors had access to information that could identify individual participants during or after data collection.

Ethical clearance was provided by the Ethical Research Committee (Faculty of Medicine University of Indonesia S-996/UN2.F1/ETIK/PPM.00.02/2019). Written informed consent was obtained from the participants' parents or guardians.

The results of the multivariate logistic regression were presented as an adjusted Odds ratio (aOR) with a 95% confidence interval (CI). Significance was set as $p < 0,05$. Transformation of scoring system and risk classification of myocardial iron overload were also carried out [9].

## Results

There were 80 children with thalassemia major living in 3 large provinces in Indonesia, namely Jakarta, Banten, and West Java. The mean age was 12.3±2.82 years old, 39 males (48.8%) and

41 females (51.2%), and no significant differences between the sexes. The mean ferritin level was 3827.78 (1002.46–17.976) ng/mL, where 9 (11.25%), 47 (58.75%), and 24 (30%) patients had values of 1000–2000 ng/mL, 2000–5000 ng/mL, and > 5.000 ng/mL, respectively.

The mean MRI T2* in this study was 32.5 (9.9–46,7) ms, and 8 (10%) of this suffered for cardiac iron overload. The youngest patients affected was 9.7 years old, while the oldest was 16.5 years old respectively.

The diagnosis of thalassemia major was based on Hb analysis, and the age of first diagnosis and transfusion varied widely. Among the samples, 56 (70%) were diagnosed before the age of 1 year.

Based on left arm circumference measurment, 29 (33.3%), 24 (27.3%), and 27 (30.7%) patients had good, bad, and poor nutritional statuses, All samples received chelation therapy at 3–6 years with different adherence. A total of 22 (27.5%), 44 (55%), and 14 (17.5%) patients received blood transfusion within 2, 3, and 4-week intervals, respectively as shown in Table 1.

Our study showed proportion children with Hb of < 8 gr/dL was quite high (14%) and 69 (86%) had Hb levels of 8–10 gr/dL. All had normal IG values, with of 8 (10%) had an increase in white blood count (>10,000/uL) and of 14 (14,8%) had hs-CRP > 3 mg/L, with no fever nor signs of infection.

Of 19 (24%) children had reticulocyte > 1.49 and of 12,5% showed. thrombocytopenia, but without a history of bleeding, petechiae, and ecchymosis. There were increasing of AST (>27 u/L), ALT (>23 u/L), SI (>170 uL), and TIBC (>450 uL) in 48 (60%), 47 (58.7%), 33 (41.3%), and 2 (2.5%) patients, respectively. Further total protein, albumin, urea, and creatinine levels were normal.

This study revealed QoTC interval of > 44 ms found in 61 (76.3%) but no abnormal rhythm, P, QRS and T wave, left or right ventricle hypertrophy.

Echocardiography showed that LVDD, EF, and TAPSE were within normal limits. We found left ventricular diastolic dysfunction in 7 (8.75%) mitral E wave >0.70 m/s, 3 (3.8%) E/A ratio, 17 (21.25%) DT <110 m/s, and 4 (5%) TR Vmax >2.8 m/s, respectively.

Tissue Doppler Ecocardiography showed basal LV systolic myocardial velocity dysfunction Sm <6.4 cm/s in 4 (5%), diastolic dysfunction Em <11.1 m/s in 1 (1.3%), and Am <4.3 m/s in 2 (2.5%) children, respectively, and none of them had E/Em ratio > 8. The basal septal systolic myocardial velocity, early diastolic, and late diastolic were within normal at 7.21±0.87 cm/s, 12.44±1.52 cm/s, and 6.19±1.31 cm/s, respectively, as shown in Table 2.

**Table 1. Baseline characteristics among children with thalassaemia major.**

| Variable (n = 80) | Results Means; median (min–max) |
|---|---|
| Age (years) | 12.3±2.8 |
| Male: female, n (%) | 39 (48.8): 41 (51.2) |
| Ferritin (ng/mL) | 3827.78 (1002.46–17.976) |
| MRI T2* (ms) | 32.5 (9.9–46.7) |
| Thalassaemia: 46major: β/HbE | 65 (81%):13 (16%) |
| Left arm circumference (cm) | 14.8 (14–24.4) |
| Age at first diagnosis (months) | 8 (2–108) |
| Age at first transfusion (months) | 8 (2–108) |
| Interval between transfusions (weeks) | 2.9 (3–4) |
| Age at first chelation therapy (years) | 1–8 |
| Type of chelation therapy: FDN: DFR, n (%) | 65 (81.3): 15 (18.7) |

DFN deferiprone; FDR deferiprox; MRI magnetic resonance imaging.

**Table 2. Laboratory, ECG, echocardiography and TDI results.**

| | Variable | Result x±SD; median (min–max) | Normal values |
|---|---|---|---|
| Laboratory | Hb (g/dL) | 9.2±1.1 | 12.0–16.0 |
| | White blood count (/uL) | 6,090 (2100–21,020) | 4,500–13,000 |
| | Immature Granulocyte | 0,07 (0,01–0,63) | ≤ 10 y: < 0,30 >10 y: 0,74 |
| | hs-CRP (mg/L) | 0,65 (0–26,2) | < 3 mg/L |
| | Platelet (/uL) | $220.10^3$ ($95–672.10^3$) | $150–450. 10^3$ |
| | Reticulocyte (%) | 0.62 (0.11–18) | 0.90–1.49 |
| | AST (u/L) | 30 (12–178) | < 27 |
| | ALT (u/L) | 32 (4.2–162) | <23 |
| | SI (u/L) | 150.1±54.12 | 50–170 |
| | TIBC (u/L) | 180 (99–625) | 250–450 |
| ECG | QoTc interval (ms) | 0.45 (0.39–0.63) | >0.44 |
| Echocardiography | LV Diastolic Dimension cm) | 4.25±0.45 | 4±0.52 |
| | EF (%) | 59.5 (55.1–78.1) | > 55 |
| | TAPSE (cm) | (2,6±0.46 cm) | |
| | E wave (m/s) | 99.56±14.61 | <0.70 |
| | DT (m/s) | 132.9±26.2 | < 110 |
| | TR Vmax (m/s) | 2.26 (0.67–3.45) | < 2.8 |
| TDI | Sm Left Basal Lateral (cm/s) | 8.25±1.40 | 8±2.4 |
| | Em Left Basal Lateral (cm/s) | 17.08±2.65 | 10.3±2.7 |
| | E/Em Left Basal Lateral Ratio | 5.94±1.03 | <8 |
| | Sm basal septal (cm/s) | 7.21±0.87 | 6.1±1,7 |
| | Em basal septal (cm/s) | 12.44±1.52 | 10.3±2.7 |

The proportion of free variables in myocardium iron overload was assessed by performing a bivariate analysis between free and dependent variables. Receiver Operating Characteristics (ROC) procedure was carried out on all free variables. The results showed that Hb, reticulocyte, AST, ferritin, chelation adherence, DT, and TR Vmax had an AUC of > 0.60 (Table 3).

The free variables associated with cardiac iron overload based on the ROC curve included Hb, reticulocyte, AST, ferritin, chelation adherence, DT, and TR Vmax, as shown in Table 4. Further, determinant factors that played an independent role in cardiac iron overload were Hb, reticulocyte, DT, and TR Vmax, as shown in Table 5.

Based on the results, the formula for cardiac iron overload was -9.32 + 3.28 (Hb) + 2.79 (Reticulocyte) + 3 .64 (DT) + 4.10 (TR Vmax). Furthermore, the Hosmer and Lameshow test revealed good calibration with p = 0.52 and as well as a good discriminant value at AUC = 0.96 (CI 95% 0.898–1.000) with Nagelkerke R square of 0.68 (Fig 1).

**Table 3. Determination of free variable intersecting point based on ROC curve.**

| Free variable | Intersecting Point | Sensitivity | Specificity | AUC (CI 95%) |
|---|---|---|---|---|
| Hb (g/L) | ≤ 8.25 | 0.86 | 0.63 | 0.67 (0.41–0.93) |
| Reticulocyte (%) | ≤0.33 | 0.74 | 0.75 | 0.74 (0.57–092) |
| AST (U/L) | ≥36.5 | 0.75 | 0.75 | 0.70 (0.52–0.87) |
| Ferritin (ng/dL) | ≥5,200 | 0.75 | 0.63 | 0.70 (0.48–0.92) |
| Chelation Adherence (%) | ≤82 | 0.75 | 0.79 | 0.77 (0.59–0.91) |
| DT (cm/s) | ≤114.5 | 0.81 | 0.63 | 0.87 (0.79–0.96) |
| TR Vmax (m/s) | ≥2.37 | 0.76 | 0.88 | 0.82 (0.67–0.96) |

**Table 4. Bivariate analysis of free variable, presence of iron overload predictor.**

| Free Variable | | Group | | | | CI 95% |
|---|---|---|---|---|---|---|
| Variable | Intersecting point | cardiac iron overload (n = 8) | No cardiac iron overload (n = 72) | p-value | OR | MM Min–Max |
| Hb (g/dL) | ≤8.25 | 5 (33.30) | 10 (66.70) | 0.00 | 10.33 | 2.13–50.14 |
| | >8.25 | 3 (4.60) | 62 (95.40) | | | |
| Reticulocyte (%) | ≤0.33 | 6 (24.00) | 19 (76.00) | 0.01 | 8.37 | 1.56–45.08 |
| | >0.33 | 2 (3.60) | 53 (96.40) | | | |
| AST (u/L) | ≥36.5 | 6 (25.00) | 18 (75.00) | 0.01 | 9.00 | 1.67–48.62 |
| | <36.5 | 2 (3.60) | 54 (96.40) | | | |
| Ferritin (ng/dL) | ≥5,200 | 5 (21.70) | 18 (78.30) | 0.04 | 5.00 | 1.09–23.03 |
| | <5,200 | 3 (5.30) | 54 (94.70) | | | |
| Chelation adherence (%) | ≤82 | 6 (28.6) | 15 (71.4) | 0.00 | 11.4 | 2.09–62.3 |
| | >82 | 2 (3.4) | 57 (96.6) | | | |
| DT (cm/s) | ≤114.5 | 5 (26,30) | 14 (73.70) | 0.02 | 6.91 | 1.47–32.39 |
| | >114.5 | 3 (4.90) | 58 (95.10) | | | |
| TR Vmax (m/s) | ≥2.37 | 7 (29.20) | 17 (70.80) | 0.00 | 22.65 | 2.60–197.31 |
| | <2.37 | 1 (1.80) | 55 (98.20) | | | |

## Formula of cardiac iron overload based on scoring system

The formula for assessing cardiac iron overload was simplified and scoring system was defined, where the smallest B/SE result (2.06) was reticulocyte with 1 point. Furthermore, Hb, DT, and TR Vmax variables were simplified by rounding up the values, as 1.19, 1.13 and 1.12 and each variable point were simplified as 1.

On transformation of scoring system for each variable category, the dichotomy value was Hb ≤8.2 g/dL = 1 point and > 8.2 g/dL = 0 point. Reticulocyte ≤0.33% = 1 and >0.33 = 0, DT ≤114.5 cm/s = 1 and >114.5 cm/s = 0, while TR Vmax ≥ 2.37 m/s = 1 point and < 2.37 = 0 point.

The discriminant and calibration scoring model was also assessed and the total variable point was treated as a free variable. Furthermore, the probability scoring system model obtained was Y = -8.76 + 3.25 (total point). Hosmer and Lameshow tests showed good calibration (p = 0.94) and discriminant (AUC = 0.96, 95% CI 0.90–1.00).

Based on scoring system and risk classification for iron cardiac overload, scores of 0 or 1, 2, and 3 or 4 were categorized as low, moderate, and high-risk, respectively, as shown in Table 6.

## Discussion

Thalassemia major is a significant health burden in Indonesia, with an increasing number of cases reported. The main cause of death was myocardial iron overload, which underscored the

**Table 5. Determinant factors for cardiac iron overload.**

| Variable | B | SE | p-value | aOR | CI 95% |
|---|---|---|---|---|---|
| | | | | | Min–Max |
| Hb | 3.28 | 1.34 | 0.01 | 26.65 | 1.92–369.14 |
| Reticulocyte | 2.79 | 1.36 | 0.04 | 16.27 | 1.14–232.33 |
| DT | 3.64 | 1.56 | 0.02 | 38.22 | 1.80–810.62 |
| TR Vmax | 4.10 | 1.77 | 0.02 | 60.27 | 1.87–1,942.56 |
| Constant | -9.32 | 2.90 | 0.01 | 0.01 | |

Adjusted OR variables of Hb, reticulocyte, AST, DT, and TR Vmax

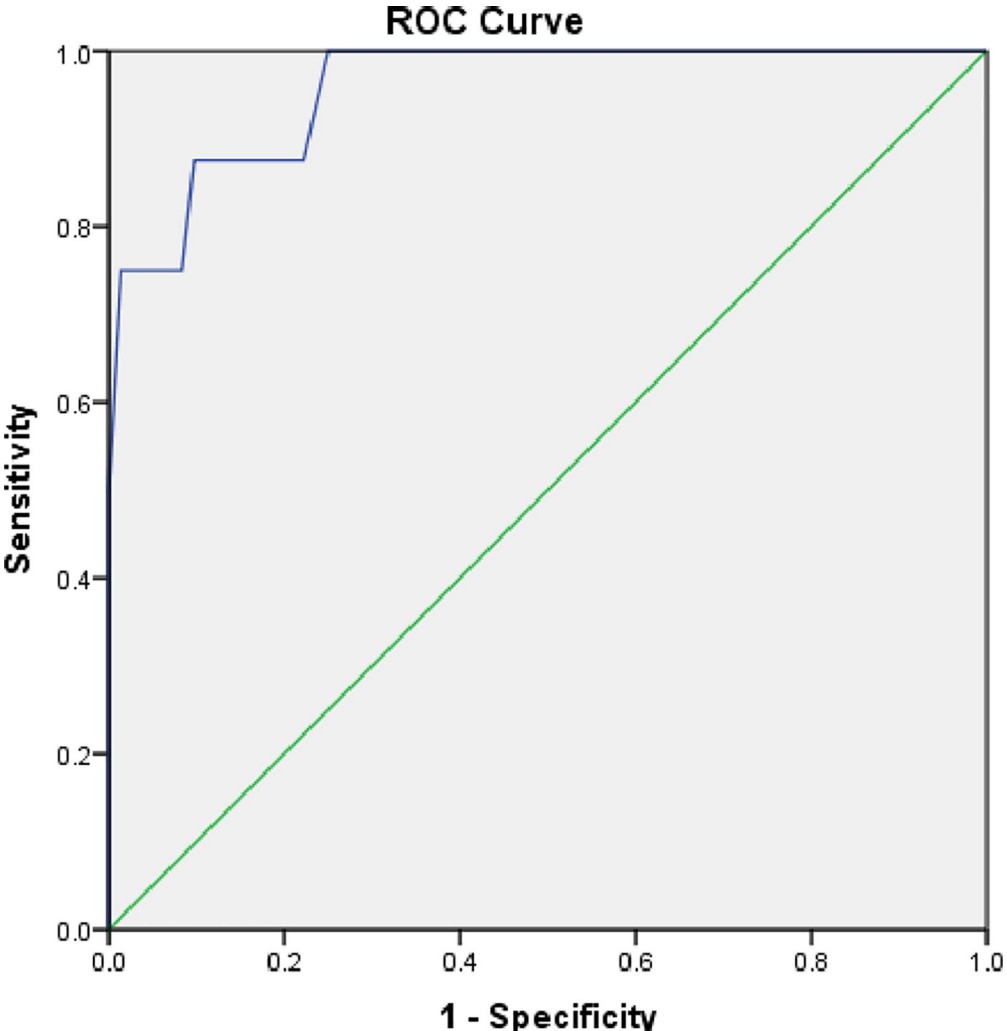

**Fig 1. ROC curve of cardiac iron overload based on formula Y = -9,32 + 3,28 (Hb) +2,79 (retikulosit) + 3,64 (DT) + 4,10 (TR Vmax).**

need for MRI T2* screening to treat and detect this condition. However, MRI T2* screening was limited due to its high costs and low availability. To address this issue, a low-cost-investigation-based formula and scoring system was needed. Based on previous findings, this is the

**Table 6. Probability and risk classification.**

| Score | Constant | Coefficient | y | Probability | Risk Classification |
|-------|----------|-------------|-------|-------------|---------------------|
| 0 | -8.76 | 3.25 | -8.76 | 0.00 | Low risk |
| 1 | -8.76 | 3.25 | -5.51 | 0.00 | Low risk |
| 2 | -8.76 | 3.25 | -2.25 | 0.10 | Moderate risk |
| 3 | -8.76 | 3.25 | 1.00 | 0.73 | High risk |
| 4 | -8.76 | 3.25 | 4.26 | 0.99 | High risk |

y = -8.76+3.25 total point

Probability = 1/(1+exp[-y])

first study in Indonesia to evaluate the use of the formula and score system for cardiac iron overload among children with thalassemia major.

We studied 80 thalassemia major with the average age of 12.3±2.8 years old, and the sex distribution was similar to that of previous studies [10]. This study recruited children aged > 6 years old who cooperated during the MRIT 2* procedure and ferritin level of > 1,000 ng/mL. As the majority of who those with ferritin level of > 1,000 ng/mL were at risk of cardiac involvement, left ventricular dilatation, abnormal contractility, tricuspid valve regurgitation, and pulmonary hypertension [11].

The mean MRI T2* range was 32.5 (9.9–46,7) ms, and this was similar to a previous study with a value of 34.4 (3.3–76) ms [12]. In this study showed proportion myocardial iron overload 8 (10%). Assis et al [13] carried out a study on 102 thalassemia aged 12 to 23 years who underwent routine transfusion with 3–4 weeks intervals, and the results showed that 36% had a myocardial iron overload. Wahidiyat et al. also assessed patients aged 14 (3–43) years and obtained a 14.8% prevalence [12]. Further, the prevalence of cardiac iron overload often differed in various countries and was affected by age, number of blood transfusions, and adherence to chelation therapy [14].

The transfusion intervals were 2.9 (3–4) weeks, among of 14 (17,5%) children had 4 weeks interval transfusions and 13.7% had Hb levels of < 8 g/dL. Parents' ignorance of the need for regular transfusion, low social economic, and living in far remote areas were among the cause of the prolonged interval and low Hb level. The 3 weeks transfusion interval aimed to increase low pre-transfusion Hb as well as maintain the level of 10 gr/dL to prevent developmental problems.

Another major contributor to the occurrence of cardiac iron overload was the lack of adherence to chelation therapy due to boredom. There was also a weak correlation between adhering to therapy and myocardial iron overload (r 0.25 p = 0.02). The odd ratio (OR) of chelation adherence of ≤ 82% for developing myocardial iron overload was 11.4 (95% CI 2.09–62.3, p = 0.00). However, the results must be carefully interpreted since the adherence method was calculated with history taking with no medical record support. This was one of limitation of this study. However the adherence to chelation therapy variable was not play a rule on the predicting formula of myocardiac iron overload.

Transfusion without chelation therapy can lead to iron deposition as well as progressive liver, endocrine gland, kidney, and heart dysfunction [5], but most patients in this study received FDN. Studies showed that DFP was associated with less myocardial iron burden and better global systolic ventricular function compared to oral deferasirox or desferrioxamine [14].

Nutritional status was assessed in this study with an upper arm circumference measurement as body weight and body surface can be affected by other variables, such as organomegaly. Based on the measurement, the nutritional status was divided into several categories, where 85–100%, 70–85%, and <70% were classified as good, bad, and poor, respectively [15]. A total of 24 (27.3%) and 27 (30.7%) participants showed bad and poor status, leading to growth delay and short stature. Adequate nutrition was essential for thalassemia as a long-term therapy modality as it helped to prevent nutritional disorders, growth delays, poor puberty development, and immune deficiency [16]. The nutritional intake recommended for patients consisted of high-calorie food, potassium, zinc, vitamins A, D, E, and low iron, while vitamin C must be reduced as it can increase iron absorption [14, 16].

The IG obtained in this study was normal and none of patients suffered from bacterial infection. Of 10 (12.5%) children had thrombocytopenia, however, there was no history of bleeding, petechiae, ecchymosis, and any other related signs. Thrombocytopenia was caused by an increase in platelet destruction and reduced thrombopoietin due to the usage of DFP chelation and hypersplenism, which has a 10% incidence rate of bleeding.

PT and aPTT were not evaluated in this study, but increased levels correlated with ferritin and were commonly found in moderate to severe iron liver deposition compared to normal or mild stages [17, 18]. Delayed PTT was caused by liver dysfunction and the activation of coagulation cascade. This condition can be treated with chronic transfusion and hemolysis-induced kallikrein esterase activity, which required the XI and XII factors [18].

Reticulocyte levels of >1.49% were found in 20 (25%) patients in this study. Furthermore, there was a significant correlation between ferritin and reticulocyte (r = -0.45 p = 0.01). An increase in reticulocyte was reported to correlate with erythropoietin activity and increased ferritin levels [19].

The increase of AST, ALT, SI, and TIBC, which was consistent with previous studies [20]. There was also a significant correlation between AST and ferritin (r = 0.27 p = 0.02), but AST (r = 0.19 p = 0.09) did not correlate with SI (r = 0.21 p = 0.85) and TIBC (r = 0.123 p = 0.275). Average ferritin levels were observed to be higher in groups with increased AST and ALT (r = 0.01 and p = 0.01) [20].

Our study did not carry out Hepatitis B and C screening. Previous reports showed that patients with these conditions had increased AST, ALT, and serum ferritin levels [21]. AST was often present in liver, heart, skeletal muscle, brain, and kidney. It can also be found in the cytosol or mitochondria, and its levels can be increased due to mitochondrial damage. Meanwhile, ALT was reported to be abundant in the liver cytoplasm and more specific in detecting liver dysfunction compared to AST [20, 21].

We found that ECG, echocardiography, and TDI variables did not correlate with MRI T2*. There was also no correlation between QoTc interval with ferritin. These results are inconsistent with those obtained in previous studies due to differences in subject characteristics [22, 23].

Conventional echocardiography showed that both ejection fraction and right systolic function were normal, but there was early LV diastolic dysfunction (E mitral wave > 0.7 m/s) in 77 (96,3%) patients. The TDI study showed lateral myocardial velocity systolic dysfunction (Sm wave) among children with normal EF, and 3 (3,75%) patients had an E/Em ratio of > 8. Systolic and diastolic myocardial velocity dysfunction on TDI has been reported in other studies [24, 25]. TDI was very useful in detecting this condition even in cases with reserve normal EF [26]. The normal basal septal myocardial velocity in this study indicated that the lateral LV wall was affected earlier than the septal part.

A significant difference in QoTc interval, E mitral wave, DT, reduction in Sm lateral wave, and Sm septal wave were found among thalassemia patients with ferritin of >5,000 ng/dL, 2,500–5,000 ng/dL, and <2,500 ng/dL. The difference in result was caused by the age of patients, who were older compared to those in this study [27]. A significant variation in ECG and echocardiography variables were found in the thalassaemia group compared to the normal group. A significant difference was also found between children and adults [28].

This study showed the determinant factors for cardiac iron overload were Hb, reticulocyte, DT, and TR Vmax. Low hemoglobin can affect the cardiovascular system, leading to haemodilution, venous dilation, decreased blood pressure, and increased LVDD. Decreasing in DT in our styudy was consistent with other studies and might followed by E mitral wave abnormality. There is a good correlation between DT, E/Em index and MRIT2* value [29].

Thalassaemia can cause pulmonary hypertension (PH) and the gold standard was cardiac catheterization. In this study, we predicted PH by performing the TR Vmax on echocardiography. We found the average of TR Vmax was 2.25 (0.7–3.5) m/s, of those 24 (30%) were TR Vmax of >2.5 m/s and were 9 (11.25%) TR Vmax >2.8 m/s, respectively. Normal thalassemia major pulmonary arterial pressure is characterized by TR Vmax of 2.5–2.7 m/s. TR Vmax

of $\geq$ 2.5 m/s can be utilized as a screening test to determine the risk for PH [30]. TR Vmax of > 3.4 m/s were significant indicators for pulmonary hypertension [31].

The mechanism of PH was multifactorial, and chronic hemolysis was related to a decrease in bioavailability of nitric oxide (NO) and platelet activation as well as an increase in adhesion molecules and vasoactive peptide Endothelin-1. Hypercoagulopathy has been reported to cause macrovascular and microvascular thrombosis, while splenectomy can lead to a coagulation activation factor [32, 33].

The formula for cardiac iron overload was -9.32+3.28 (Hb) + 2.79 (reticulocyte) + 3,64 (DT)+ 4.1 (TR Vmax). The Hosmer and Lameshow test showed good calibration. A scoring system was then defined by simplifying the formula. Values of Hb $\leq$ 8.2 g/L, reticulocyte $\leq$0.33%, DT $\leq$ 114.5 cm/s, and TR Vmax $\geq$ 2.37 m/s were given a score of 1, while others were assigned 0. Subsequently, risk for iron cardiac overload, total scores of 0 or 1, 2, and 3 or 4 were categorized as low, moderate, and high, respectively.

The formula and scoring system were developed to improve management in monitoring cardiac iron overload as the leading cause of death among children which is prevalent. Scoring system can be used as an alternative assessment tool in the absence of MRI T2*, thereby increasing patient survival especially in regions where MRI T2 is unavailable as our country as well as other low- and middle-income countries. It may lighten the economic burden of thalassemia management cost. Additionally, the limitations of the study might present opportunities for future research endeavors.

## Conclusions

The formula for cardiac iron overload in this study was -9.32+ 3.28 (Hb) + 2.9 (reticulocyte) + 3.64 (DT) + 4.1 (TR Vmax). Variables of Hb $\leq$ 8.2 g/L, reticulocyte $\leq$0.33%, DT $\leq$ 114.5 cm/s, and TR Vmax $\geq$ 2.37 m/s were given a score of 1, while others were assigned 0. Based on the scoring system and risk classification, values of 0 or 1, 2, and 3 or 4 were categorized as low, moderate, and high-risk for cardiac iron overload, respectively.

## Supporting information

**S1 Checklist.** *PLOS ONE* **clinical studies checklist.**
(DOCX)

**S2 Checklist. STROBE statement—Checklist of items that should be included in reports of observational studies.**
(DOCX)

**S1 File.**
(XLSX)

**S2 File.**
(PDF)

**S3 File.**
(PDF)

**S4 File.**
(XLSX)

## Acknowledgments

The authors are grateful to Prof. Elizabeth Nemeth for providing advice and expert opinion.

## Author Contributions

**Conceptualization:** Syarif Rohimi, Bambang Budi Siswanto.

**Data curation:** Syarif Rohimi, Muchtaruddin Mansyur, Ina Sutanto.

**Formal analysis:** Syarif Rohimi, Muchtaruddin Mansyur.

**Funding acquisition:** Syarif Rohimi.

**Investigation:** Syarif Rohimi.

**Methodology:** Muchtaruddin Mansyur.

**Resources:** Djajadiman Gatot.

**Supervision:** Bambang Budi Siswanto, Djajadiman Gatot, Ina Sutanto, Jacub Pandelaki.

**Validation:** Bambang Budi Siswanto, Ina Sutanto, Jacub Pandelaki, Amiliana M. Soesanto, Teddy Ontoseno.

**Visualization:** Djajadiman Gatot, Jacub Pandelaki, Amiliana M. Soesanto, Teddy Ontoseno.

**Writing – original draft:** Syarif Rohimi.

**Writing – review & editing:** Bambang Budi Siswanto, Muchtaruddin Mansyur, Djajadiman Gatot, Ina Sutanto, Amiliana M. Soesanto, Teddy Ontoseno.

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
