## [Decision Letter · Decision Letter 0]

2 Apr 2024

PONE-D-23-18085The Predicting formula and scoring system for cardiac iron overload for thalassemia children:  an urgent need in IndonesiaPLOS ONE

Dear Dr. Rohimi,

Thank you for submitting your manuscript to PLOS ONE. After careful consideration, we feel that it has merit but does not fully meet PLOS ONE’s publication criteria as it currently stands. Therefore, we invite you to submit a revised version of the manuscript that addresses the points raised during the review process.

**ACADEMIC EDITOR: **

It is undoubtedly a very good paper, so I urge the authors to consider the indications of the reviewers, especially, to deepen the discussion and the projections of the article.

We look forward to receiving your revised manuscript.

Kind regards,

Juan Luis Castillo-Navarrete, Ph.D.

Academic Editor

PLOS ONE

Reviewers' comments:

Reviewer's Responses to Questions

**Comments to the Author**

1. Is the manuscript technically sound, and do the data support the conclusions?

Reviewer #1: Yes

Reviewer #2: Yes

2. Has the statistical analysis been performed appropriately and rigorously? 

Reviewer #1: Yes

Reviewer #2: Yes

3. Have the authors made all data underlying the findings in their manuscript fully available?

Reviewer #1: No

Reviewer #2: Yes

4. Is the manuscript presented in an intelligible fashion and written in standard English?

Reviewer #1: Yes

Reviewer #2: Yes

5. Review Comments to the Author

Reviewer #1: According to what was presented, conclusive data are obtained to estimate a score that correlates with iron overload. A correlation is also suggested between adherence to chelation treatment and number of transfusions, as it may bias the results.

Detailed demographic data is not found in the supplementary material. It is suggested to attach them to review the data tabulation.

Reviewer #2: Dear Authors,

I hope this letter finds you well. I am delighted to have the opportunity to review your exceptional research in the field of Thalassemia. Your findings are invaluable contributions to the scientific community and hold great significance for both researchers and patients alike. Please allow me to provide some constructive feedback aimed at enhancing the quality of your manuscript, organized according to each section:

1. Title: The title is informative and relevant, effectively capturing the essence of your study.

2. Abstract: I would kindly suggest mentioning the formula in the conclusion section of the abstract. Additionally, incorporating relevant keywords such as "MRI T2" could prove beneficial for indexing and discoverability purposes. Furthermore, I recommend replacing the term "thalassemia major" with the more encompassing "thalassemia" throughout the abstract. The remaining portions of the abstract effectively convey the study's findings and methodological approach.

3. Introduction: While conciseness is appreciated, a more substantial introduction could better communicate the importance and necessity of your research. The research question could be more firmly grounded in the existing literature, necessitating the inclusion of more up-to-date studies and references, particularly recently published works such as those by Meloni et al. and Fedai et al.

4. Methods: The subject selection process is clearly delineated. The variables are well-defined and measured appropriately, and the study methods are valid and reliable. The level of detail provided is sufficient to facilitate replication of the study by others.

5. Results: The presentation of data could be enhanced through the judicious use of visual aids, such as charts and graphs, which would effectively communicate the key findings of your research. A Receiver Operating Characteristic (ROC) graph is highly recommended. Appropriate labeling of columns and rows in tables is appreciated, but I would suggest removing any horizontal lines to enhance readability.

6. Discussion and Conclusions: A more comprehensive discussion, exploring the results from multiple perspectives, would be beneficial. Addressing the socioeconomic implications of your findings and the potential of the formula for early detection of cardiac iron overload in regions where MRI T2 is unavailable could broaden the impact of your work. Generalizing your results to low- and middle-income countries, where the problem you have identified is prevalent, would further underscore the global relevance of your research. The conclusions are well-supported by the results and effectively address the aims of the study. Additionally, the limitations acknowledged present opportunities for future research endeavors.

7. Overall: The study design is appropriate and effectively addresses the research aims, contributing novel insights to the existing body of knowledge on this topic.

I commend your exceptional work and look forward to the positive impact it will have on the scientific community and the lives of those affected by Thalassemia.

6. PLOS authors have the option to publish the peer review history of their article (what does this mean?). If published, this will include your full peer review and any attached files.

Reviewer #1: **Yes: **Juan Carlos Rivera Fuentes

Reviewer #2: **Yes: **Amirreza Nasirzadeh

---

## [Author Response · Author response to Decision Letter 0]

10 Jul 2024

Respons to Reviewers

ACADEMIC EDITOR: 

It is undoubtedly a very good paper, so I urge the authors to consider the indications of the reviewers, especially, to deepen the discussion and the projections of the article.

Done

Done

Done

No changes on financial disclosure

Revision: laboratory protocols has been added

We look forward to receiving your revised manuscript.

Kind regards,

Juan Luis Castillo-Navarrete, Ph.D.

Academic Editor

PLOS ONE

Revision: Abstract structure has been revised

Reviewers' comments:

Reviewer's Responses to Questions

Comments to the Author

1. Is the manuscript technically sound, and do the data support the conclusions?

Reviewer #1: Yes

Reviewer #2: Yes

2. Has the statistical analysis been performed appropriately and rigorously?

Reviewer #1: Yes

Reviewer #2: Yes

3. Have the authors made all data underlying the findings in their manuscript fully available?

Reviewer #1: No

Reviewer #2: Yes

Revision: means data has been added

4. Is the manuscript presented in an intelligible fashion and written in standard English?

Reviewer #1: Yes

Reviewer #2: Yes

5. Review Comments to the Author

Reviewer #1: According to what was presented, conclusive data are obtained to estimate a score that correlates with iron overload. A correlation is also suggested between adherence to chelation treatment and number of transfusions, as it may bias the results.

Detailed demographic data is not found in the supplementary material. It is suggested to attach them to review the data tabulation.

Explanation: During study, the chelation adherence and number of transfusion data were taken with history taking as most subjects didn’t have adequate monitoring book so might be bias. However in this study the chelation adherence and number of transfusion were not a parameter for making a formula

Reviewer #2: Dear Authors,

I hope this letter finds you well. I am delighted to have the opportunity to review your exceptional research in the field of Thalassemia. Your findings are invaluable contributions to the scientific community and hold great significance for both researchers and patients alike. Please allow me to provide some constructive feedback aimed at enhancing the quality of your manuscript, organized according to each section:

1. Title: The title is informative and relevant, effectively capturing the essence of your study.

Revision: The predicting formula and scoring system for cardiac iron overload for thalassaemia children: Study from a Middle-Income Country 

Short title: A formula and scoring system for cardiac iron overload in thalassemia children

As a suggestion on Discussion and Conclusions below: Addressing the socioeconomic implications of your findings and the potential of the formula for early detection of cardiac iron overload in regions where MRI T2 is unavailable could broaden the impact of your work.

2. Abstract: I would kindly suggest mentioning the formula in the conclusion section of the abstract. Additionally, incorporating relevant keywords such as "MRI T2" could prove beneficial for indexing and discoverability purposes. Furthermore, I recommend replacing the term "thalassemia major" with the more encompassing "thalassemia" throughout the abstract. The remaining portions of the abstract effectively convey the study's findings and methodological approach.

Revision: I have put the formula in the conclusion section of the abstract. I have also replaced the term of thalassemia major with thalassemia throughout the abstract.

3. Introduction: While conciseness is appreciated, a more substantial introduction could better communicate the importance and necessity of your research. The research question could be more firmly grounded in the existing literature, necessitating the inclusion of more up-to-date studies and references, particularly recently published works such as those by Meloni et al. and Fedai et al.

Revision: to a better communicate the importance and necessity of the study, I have added some information regarding side effect of regular blood transfusion such as cardiac fibrosis and pulmonary hypertension on introduction. Regular blood transfusions are required to treat thalassemia and might cause iron overload, organ dysfunction, heart failure (HF), cardiac fibrosis, pulmonary hypertension and death [1,3,5].

I have revised literature:

no 31 with a recently published work which done by Sharma M, Burns AT, Yap K, Prior DL. The role of imaging in pulmonary hypertension. Cardiovasc Diagn Ther 2021; 11:859-80. No 32. Fraidenburg DR, Machado RF. Pulmonary hypertension associated with thalassemia syndromes. Ann N Y Acad Sci. 2016;1:127–39.

4. Methods: The subject selection process is clearly delineated. The variables are well-defined and measured appropriately, and the study methods are valid and reliable. The level of detail provided is sufficient to facilitate replication of the study by others.

No revision done

5. Results: The presentation of data could be enhanced through the judicious use of visual aids, such as charts and graphs, which would effectively communicate the key findings of your research. A Receiver Operating Characteristic (ROC) graph is highly recommended. Appropriate labeling of columns and rows in tables is appreciated, but I would suggest removing any horizontal lines to enhance readability.

Revision: a ROC graph have been added on the manuscript

6. Discussion and Conclusions: A more comprehensive discussion, exploring the results from multiple perspectives, would be beneficial. Addressing the socioeconomic implications of your findings and the potential of the formula for early detection of cardiac iron overload in regions where MRI T2 is unavailable could broaden the impact of your work. Generalizing your results to low- and middle-income countries, where the problem you have identified is prevalent, would further underscore the global relevance of your research. The conclusions are well-supported by the results and effectively address the aims of the study. Additionally, the limitations acknowledged present opportunities for future research endeavors.

Revision: Addressing the socioeconomic implication has been added to the manuscript as well as the possibility of using formula and score system for the others low and middle income countries

7. Overall: The study design is appropriate and effectively addresses the research aims, contributing novel insights to the existing body of knowledge on this topic.

I commend your exceptional work and look forward to the positive impact it will have on the scientific community and the lives of those affected by Thalassemia.

6. PLOS authors have the option to publish the peer review history of their article (what does this mean?). If published, this will include your full peer review and any attached files.

---

## [Decision Letter · Decision Letter 1]

16 Aug 2024

The Predicting formula and scoring system for cardiac iron overload for thalassemia children:  study from a middle income country

PONE-D-23-18085R1

Dear Dr. Syarif Rohimi,

We’re pleased to inform you that your manuscript has been judged scientifically suitable for publication and will be formally accepted for publication once it meets all outstanding technical requirements.

Kind regards,

Juan Luis Castillo-Navarrete, Ph.D.

Academic Editor

PLOS ONE

Additional Editor Comments (optional):

Congratulations to the authors for the genesis of an excellent piece of writing.

Reviewers' comments:

Reviewer's Responses to Questions

**Comments to the Author**

1. If the authors have adequately addressed your comments raised in a previous round of review and you feel that this manuscript is now acceptable for publication, you may indicate that here to bypass the “Comments to the Author” section, enter your conflict of interest statement in the “Confidential to Editor” section, and submit your "Accept" recommendation.

Reviewer #1: (No Response)

Reviewer #2: All comments have been addressed

2. Is the manuscript technically sound, and do the data support the conclusions?

Reviewer #1: Yes

Reviewer #2: Yes

3. Has the statistical analysis been performed appropriately and rigorously? 

Reviewer #1: Yes

Reviewer #2: Yes

4. Have the authors made all data underlying the findings in their manuscript fully available?

Reviewer #1: No

Reviewer #2: Yes

5. Is the manuscript presented in an intelligible fashion and written in standard English?

Reviewer #1: Yes

Reviewer #2: Yes

6. Review Comments to the Author

Reviewer #1: I would like to start by greeting the authors.

I suggest adding a table that organizes both the application of the formula along with the risk ranges of a simpler one so that clinicians can apply it. The constructed tables show the results obtained and validated, but one that simplifies their application is missing.

In the review they mention the preparation of OR graphs but they do not appear in the article. I suggest adding them.

When consulting the bibliography, an adequate focus on the topic in question is evident. I suggest updating it and strengthening the quotes used.

My most sincere congratulations for the work done.

Reviewer #2: Dear authors,

Thank you for applying the suggested comments. The present version of the manuscript is suitable for publication in PLOS ONE.

7. PLOS authors have the option to publish the peer review history of their article (what does this mean?). If published, this will include your full peer review and any attached files.

Reviewer #1: **Yes: **Juan Carlos Rivera Fuentes

Reviewer #2: **Yes: **Amirreza Nasirzadeh

---

## [Editor Report · Acceptance letter]

26 Aug 2024

PONE-D-23-18085R1 

PLOS ONE

Dear Dr. Rohimi, 

I'm pleased to inform you that your manuscript has been deemed suitable for publication in PLOS ONE. Congratulations! Your manuscript is now being handed over to our production team.

Kind regards, 

on behalf of

Dr. Juan Luis Castillo-Navarrete 

Academic Editor

PLOS ONE